# Bone Mineral Density in Adolescent Boys: Cross-Sectional Observational Study

**DOI:** 10.3390/ijerph18010245

**Published:** 2020-12-31

**Authors:** Anna Kopiczko, Jakub Grzegorz Adamczyk, Monika Łopuszańska-Dawid

**Affiliations:** 1Department of Human Biology, Faculty of Physical Education, Józef Piłsudski University of Physical Education in Warsaw, 00-968 Warsaw, Poland; anna.kopiczko@awf.edu.pl (A.K.); monika.lopuszanska@awf.edu.pl (M.Ł.-D.); 2Department of Theory of Sport, Faculty of Physical Education, Józef Piłsudski University of Physical Education in Warsaw, 00-968 Warsaw, Poland

**Keywords:** bone mineral content, physical activity, birth weight, breastfeeding, fat mass, fat free mass

## Abstract

Physical inactivity of children can be a precursor of reduced bone mineral density, considered to be a typical problem only in old age. The aim of this study was to evaluate bone mineral density in 96 Polish boys aged 14–17 years with varied physical activity (swimmers, track and field athletes, non-athletes) and the effect of bone composition, birth weight and breastfeeding during infancy on bone parameters. Anthropometric and body composition measurements were performed according to the kinanthropometric standards. Bone parameters of the forearm were measured by means of dual-energy X-ray absorptiometry. Data on the infant’s birth weight and the length of breastfeeding were collected during direct interviews with mothers. The strongest links with bone parameters were found for the type of physical activity and birth weight. Regardless of birth weight, track and field athletes had the most advantageous bone parameters (mainly sT-score prox values). Swimmers with normal or low birth weight had less favourable sT-score prox values than non-athletes. The type of physical activity proved to be an important determinant of bone parameters. Childhood and adolescence are important periods of bone development and increasing the content of bone mineral components, and the bone status in later years of life depends to a large extent on this period. The perinatal period, especially the correct birth weight of the child, not only has a significant effect on general health, but also on bone status.

## 1. Introduction

Childhood and adolescence are periods of progressive ontogenesis, when the skeleton microarchitecture and mineralization undergo substantial changes. It is estimated that more than half of peak bone mass (PBM) is acquired in teenage age. PBM is usually reached in the third decade of life and, if it is low, it is the predictor of osteoporosis at a later age [1]. The human skeleton initially consists of a soft fibrous material called cartilage. Then, it is gradually transformed into the bone through a process called bone mineral density (BMD) [2]. This process is determined by both non-modifiable genetic factors [3,4] and a number of modifiable, demographic, socio-economic, hormonal, and especially lifestyle-related factors [5,6,7,8]. Although genetics plays an important role in determining PBM, environmental factors such as diets [9] and level of physical activity (PA) [10], especially in late childhood and early adolescence, are considered important modulators of individual genetic potential.

The effects of nutrition and dietary supplementation on bone health are well documented, especially with regard to calcium and vitamin D [11,12]. A non-nutritional source of vitamin D obtained from the effective exposure to sunlight should be emphasized as it is ensured by sufficiently long PA in the open air [13,14].

The relationships between bone health and body composition are sometimes contradictory [15]. In the adult population, BMD often shows a stronger correlation with the body mass index (BMI) and lean body mass, and much smaller with body fat mass [16,17]. Some studies of healthy mature adolescents and young adults have shown that high levels of adipose tissue are not beneficial for bone structure [18] or show no significant relationship [12]. The studies have demonstrated that fat mass was inversely proportional to bone mineral content (BMC) after removal of the mechanical load effect [19]. Furthermore, in children in the prepubescent period, a relationship between fat tissue and bone growth stimulation has been sought and a positive relationship between fat mass and bone mass was established [20]. Low PA was associated with a higher chance of overweight and central obesity [21]. As the influence of adipose tissue on BMD is not clear and the PA in adolescents promotes lower body fat [22], searching for the effect of specific activities on bone status seems to be justified. Birth variables are also considered from the standpoint of their relationship with bone status during progressive ontogenesis. The lower PBM and higher incidence of osteopenia and osteoporosis, thus a significantly increased risk of future fractures, were found to be significantly determined by the declared low birth weight (LBW) [23]. Further evidence suggests that LBW is linked to a subnormal PBM peak, but the mechanism of this effect needs further research [2,24].

Some studies have evaluated the effects of the exposure to nutrients in the first year of life on BMC and BMD in individuals of different ages. Breastfeeding is the recommended way of feeding infants. The results of research on the short- and long-term effects of the method of infant nutrition on BMC and density are ambiguous. Some studies have shown positive effects of breastfeeding on BMC during childhood and adolescence [25,26]. On the other hand, other publications failed to demonstrate significant relationships between the length of breastfeeding and BMC [27,28]. Interestingly, one study of adults showed the negative effect of too long breastfeeding time during infancy, on BMC in adults men. No such dependence was noted among women. [29]. In the light of the literature, there is no consensus on the effect of the method of infant feeding on BMC of people of different ages.

Bone modelling is sensitive to mechanical loads [30], thus the importance of PA during the period of bone growth should be stressed. Studies have found that physical exercise has a positive effect on BMD. However, most experimental trials have been carried out in groups of women of menopausal age to assess the risk of osteoporosis [10,31]. Cross-sectional studies show that the ways of performing exercises requiring substantial forces that generate high impacts have the highest osteogenic potential [32]. However, at the age of progressive growth, children and young people may respond differently to similar mechanical stress. The human body of physically inactive children may respond to small loads to improve BMC and structure, whereas more active children will require higher mechanical loads to obtain bone responses [33].

Physically active children and adolescents usually have better mineralization than their physically inactive peers [1]. However, not all training methods have demonstrated positive effects on BMC. For example, unloaded exercises such as swimming often have no effect on BMC [34], while walking or running has a limited positive effect [32]. However, researchers agree that it still needs to be clarified which types of PA in combination with other factors such as body composition have the greatest positive effect on bone health.

Sedentary lifestyles and abnormal body composition are the main causes of childhood obesity and related chronic diseases. However, childhood inactivity has been rarely mentioned as a precursor of osteoporosis, a disease considered to be typical only of old age. Therefore, the aim of the study was to assess BMC and BMD in young boys with different types of PA and to evaluate the effects of body composition, birth weight, and length of breastfeeding during infancy on bone parameters.

## 2. Material and Methods

### 2.1. Study Population

The study group in this cross-sectional observational study consisted of 96 young Polish boys aged 14–17 years (mean 16.11 years, SD = 1.10 years) with different types of undertaken PA. All boys were of the same ethnic origin (European). We’ve used a deliberate random model of group selection, as we have intentionally selected extremely different sports, and then the invitation for participation in the study was sent to randomly chosen clubs (athletes) from Mazovian region specialised in both track & field and swimming as well as schools (untrained peers). The study group was divided into three subgroups depending on the type of PA: swimmers (30 boys with training experience of 6.5 ± 2.4 years), track and field athletes (32 boys with training experience of 4.6 ± 1.2 years) and non-athletes (34 boys with normal daily PA-only physical education lesson 4 h/week at school). The study was attended by trained boys with training internship exclusively in their sport. Trainee groups had 4–5 training units per week. The study was carried out from 26 September to 2 December 2019 on working days from Monday to Friday in the morning. Parents gave consent for their children’s participation in this study. All young boys included in the study, and theirs parents were informed about the aims and schedule of the study. The study involved boys who were invited, and did not have the diseases described in the exclusion criteria. The exclusion criteria included bone metabolic diseases, kidney disease, thyroid and parathyroid diseases, cancers, rheumatoid arthritis, and long-term steroid treatment. The study included boys who, according to their mother’s interview, were assessed as healthy full-term newborns (i.e., born between 38th and 42nd week of pregnancy).

The measurements were conducted in the Department of Human Biology in Anthropology section, the Józef Piłsudski University of Physical Education in Warsaw, Poland, in the laboratory of densitometry and anthropometric tests. The team with the necessary qualifications and experience in research performed the measurements on the entire study group.

### 2.2. Ethical Approval

The research was carried out in accordance with the Code of Ethics of the World Medical Association (Declaration of Helsinki) for experiments involving humans. The project was approved by the Senate Ethics Committee for Scientific Research of the Józef Piłsudski University of Physical Education in Warsaw (protocol number 01–09/2017).

### 2.3. Measurements

#### 2.3.1. Somatic Measurements, Body Composition and Birth Factors

Anthropometric and body composition assessments were performed according to the International Society for the Advancement of Kinanthropometry standards [35]. Body mass and body composition were measured using a JAWON MEDICAL X-SCAN PLUS II analyzer (Certificate No. EC0197 for medical devices), with subjects barefoot and wearing light clothing. Body height was measured to the nearest 0.1 cm using the Martin Anthropometer (GMP, Switzerland). The fat percentage classification was applied according to Gallagher et al. [36] recommendation taking into account age, gender, and ethnic group.

Data on the infant’s birth weight (in grams) and the length of breastfeeding (in months) were collected during direct interviews with mothers. The study included boys who, according to their mother’s interview, were assessed as healthy full-term newborns (i.e., born between 38th and 42nd week of pregnancy).

The length of breastfeeding was evaluated to meet the criteria of the global public health recommendation by the World Health Organization [37] which states that infants should be exclusively breastfed for the first 6 months after birth to achieve optimal growth, development, and health. The birth weight of boys was compared with WHO standards [38]. LBW is defined as a birth weight of less than 2500 g (up to and including 2499). LBW is further categorized into very low birth weight (<1500 g) and extremely low birth weight (<1000 g). Large for gestational age (LGA) is used to describe newborn babies who weigh more than usual for the number of weeks of pregnancy. Babies may be called large for gestational age if they weigh more than 9 in 10 babies (90th percentile) [38].

#### 2.3.2. Bone Tissue Measurement Method

BMC and BMD of the non-dominant forearm in distal (dis) and proximal (prox) part were measured by means of dual-energy X-ray absorptiometry (DXA, Norland, Swissray, Fort Atkinson, WI, USA) using paediatric software (Norland, Warsaw, Poland). The data analysis was based on Z-scores (Z-scores are derived by comparison to a reference population on a standard deviation scale derived from an age-matched reference population) and % age-matched (and race- and gender-matched). Z-score values form the basis for the interpretation of the results of the BMD scan [39]. All measurements were taken and analysed by the same person qualified for paediatric measurements. The daily quality control and calibration of the equipment were carried out. The coefficient of variation was not determined because it was considered unethical to measure a child several times. A highly collimated beam of photons with low energy 125 I (27.4 keV) and its corresponding scintillation detector are scanned across the entire forearm at opposite sides (one-third of the distance) from a Latin stylion to an olecranon point. The total BMC depends on the size of the forearm. Absorptiometry measurements of BMC are very accurate (error of 1% to 3%).

#### 2.3.3. Statistical Analysis

Means and standard deviations were calculated for each biological parameters and for each of the three PA groups. The assumptions for the analysis of variance have not been met (e.g., normality, homoscedasticity) so the significance of differences in biological variables between PA groups was assessed using the Kruskal–Wallis test. Differences were considered significant if p was less than 0.05. The following levels of significance were used in the analyses: * *p* < 0.05; ** *p* < 0.01; *** *p* < 0.001 (*p*: *p*-value). Multiple comparisons of mean ranks for all samples were used to determine the significant differences in somatic and body composition, information on birth weight and length of breastfeeding and BMC between PA groups (track and field athletes, swimmers, non-athletes). Chi-square was used to test the significance of three PA types differences in the incidence of low BMD, low fat in body, LBW and not recommended length of breastfeeding. ANCOVA analysis of covariance with age as covariates was used to assess the strength of relationships of major determinants of biological bone mineralization status with all bone parameters. The values of adjusted determination coefficients R^2 were given. Two-way analysis of variance was employed to assess the strength of the relationships of the strongest two bone status determinants with the most ecosensitive bone parameter. In order to eliminate a significant effect of age on the determined variable, this characteristic was standardized for age with consideration of outliers.

The bone mineral variables for three PA groups or birth weight were compared to age-specific norms in accordance with the method described by Borkan and Norris [40] and often used and described in similar types of research [41,42,43]. The first step in estimating biological profile is constructing a simple piecewise linear regression equation for each trait as a function of calendar age. Then the values for each individual are compared to the reference values to determine biological age for each trait. Comparison among traits was made by standardization of residual scores using the z-transformation for the separate traits in terms of standard deviations from the mean value. The use of standardized scores allows the simultaneous representation of all of the biological parameters on a single profile graph. The significance of differences between the means for each trait was determined using the Mann–Whitney U test. Data were analysed using the STATISTICA 12.0 software package [44].

## 3. Results

The basic characteristics of the three PA groups (track and field athletes, swimmers, non-athletes) of somatic and body composition, information on birth weight and length of breastfeeding and BMC and the significance of differences are presented in Table 1. The groups differed significantly in 18 of 19 analysed biological parameters, except for body height, with no significant differences found. The non-athletes were slightly older, heavier, had higher BMI, greater fat tissue, and lower birth weight, and were the shortest breastfed compared to track and field athletes and swimmers. Compared to the other two groups, the swimmers had the least favourable parameters of bone mineralization status in seven of 10 analysed parameters, especially in the proximal segment (BMD, BMC, T-score, Z-score, % age matched). The values of bone parameters in non-athletes were between those obtained for track and field athletes and swimmers, except for Z-score dis and % age matched dis, which were the least favourable of all participants.

Table 2 shows an assessment of the incidence of low BMD (osteopenia), low body fat, LBW, and non-recommended length of breastfeeding based on the current recommendations. The highest frequency of reduced BMD occurs in swimmers, followed by the group of non-athletes. Furthermore, the most favourable body fat was recorded in swimmers whereas the most unfavourable values were found in non-athletes, with 23.5% being overweight. The incidence of each birth weight category was the most favourable for track and field athletes, while it remained at a similar level for the other two groups. Length of breastfeeding most often reached the level recommended in athletes, whereas in the other two groups, it was similarly less favourable. Approximately 67% of swimmers and non-athletes were breastfed too shortly.

The biological parameters for which the strongest links with BMC were established (i.e., age, % fat among somatic and body composition, and breastfeeding) were used for further analyses. Birth weight was also included in the analyses, because although the strength of the relationship was obtained at the level of *p* = 0.013, the literature data indicate that it may be an important predictor of bone mineralization status [45,46]. As the research by Christoffersen at al. show [46] in boys, birth weight was positively associated with BMC, standardized β coefficients (95% CI) were 0.10 (0.01, 0.19), 0.12 (0.03, 0.21) and 0.15 (0.07, 0.24) for femoral neck, total hip and total body, respectively [46]. The results of analyses of the relationships of the characteristics studied (PA, % fat, birth weight, breastfeeding, age as a continuous variable) with individual parameters of bone mineralization status (BMD, BMC, T-score, Z-score, % age matched) separately for dis and prox segments are presented in Table 3 (ANCOVA). Of all the variables analysed, the strongest relationships with bone parameters were consistently found for PA, birth weight, and age. The strength of bone status relationships with PA is mostly higher than that of the relationships with birth weight, as shown by the corresponding F test values. The T-score parameter proved to be the most reliable (for both dis and prox) and the highest values of R^2 corr were found, ranging at 15–25% of the analysed set of traits exhausting the variance of this parameter. Breastfeeding proved to be relevant only for the percentage age matched prox and Z-score prox.

The two-way analysis of variance ANOVA was also employed to provide an in-depth analysis of the observed relationships. Table 4 presents the results of the analysis of variance of parameters most strongly related to bone tissue, i.e., PA and birth weight, with the most sensitive feature, both in the dis and prox segments, i.e., T-score. In order to eliminate the significant effect of age on bone parameter status and the T-score prox variable, this feature was standardized for age (sT-score prox). Both PA and birth weight were found to be highly significantly related to the sT-score prox at age control (*p* < 0.0001), with the F-test values for PA being significantly higher and therefore the relationship being stronger. Figure 1 presents a graphical representation of the results of the analysis of variance. Regardless of birth weight, track and field athletes had the most advantageous values of sT-score prox. Swimmers with normal or LBW had less favourable sT-score prox values than non-athletes. Only large birth weight, regardless of PA, seems to guarantee a high sT-score prox.

The biological profiles for the individual bone mineral parameters, separately for the dis and prox segments for the three PA groups, are presented in Figure 2. Statistically significant differences between the average values of bone characteristics of three PA groups were found for all analysed bone parameters. In 9 for 10 parameters, these differences are at the level of *** *p* ≤ 0.001. The biological profile for athletes is clearly different from the other two as it is shifted towards more favourable values, which indicates significantly BMD. Biological profiles of swimmers and non-athletes are slightly similar and indicate less favourable values of bone parameters compared to track and field athletes. Definitely, the least favourable biological profile in seven of 10 analysed bone parameters was found in swimmers. Compared to the other two groups regardless of age, swimmers had a significantly worse bone status. The non-athlete profile in terms of Z-score dis and prox and percentage age matched dis proved to be the least advantageous compared to the other groups.

Figure 3 shows the biological profiles of the bone mineralization status of the three groups in terms of birth weight (low, normal, large). Biological profiles of the bone mineralization status for the three birth weight groups are statistically significantly different for all analysed characteristics (level of at least *p* < 0.05). Bone parameters of the participants with LBW significantly differ from those reported in the other two groups. The participants with low body weight had significantly less favourable biological bone profiles and lower bone mineralization in all analysed bone parameters. The subjects with normal and large birth weight showed significantly higher bone mineralization and more favourable profiles. Furthermore, stronger relationships were found for the prox segment. Large birth weight was associated with higher body mineralization in six of 10 analysed parameters compared to people with normal and LBW.

## 4. Discussion

The study examined bone status in young boys and demonstrated the strength and directions of the relationships of bone parameters with varied PA, body composition parameters, and birth weight and length of breastfeeding during infancy. Cases of osteopenia were reported among young boys with varied PA. Low BMD in the distal segment was found among non-athlete boys and swimmers (26.5% vs. 53.3%). In the proximal segment, a high percentage of low BMD is particularly worrying also in the same groups of boys (in 70.6% of non-athletes and 83.3% of swimmers). Among children [47], young adults [14,17,48], and physically inactive adults and older adults [6,10], the frequency of low BMD is higher than in peers undertaking regular PA. On the other hand, not all types of physical exercise and forms of movement have shown positive effects on BMD and BMC. Swimming training improves mental health parameters, cognitive abilities and motor coordination of children and adolescents [49], but at the same time does not produce osteogenic benefits [34]. The high incidence of low BMD was found, especially in the prox segment, as confirmed in swimmers. The research showed that the methods of performing and the type of exercises that require the use of high strength, generating high loads (exercises such as jumping, running, throwing, lifting, grappling, striking) have the greatest osteogenic effects, but to a different extent, on individual parts of the skeleton [50].

Examinations of total BMD and local density (spine, upper limbs, lower limbs) showed that cycling and swimming were characterized by low total BMD (1.22 and 1.17 g/cm^2^) and low lower limb BMD (1.37 and 1.31 g/cm^2^). Rugby players, soccer players, combat sports athletes, and runners had high total BMD (1.27–1.35 g/cm^2^) and high lower limb BMD (1.41–1.5 g/cm^2^). These data indicate specific local adaptations of the skeleton, as soccer players or runners improved mineralization of the lower limb bones, whereas sports based on the greater shoulder and upper body activity, such as bodybuilding, sport climbing or swimming, favour higher shoulder bone parameters [50].

In our study, we analysed the quantitative parameters of forearm bones, which would indicate the predominance of activity of these body parts in swimmers. Surprisingly, in this group of boys, the lowest BMD and BMC were recorded in both segments measured compared to track and field athletes and non-athletes. Track and field athletes had significantly better bone status, as evidenced by the higher all quantitative parameters of forearm bones. Track and field training is based on several different exercises that comprehensively develop motor skills [51] and is carried out under an axial load to the skeleton, which affects the stimulation of bones for growth and is characterized by a large osteogenic index (OI). The osteogenic potential of track and field exercises can be enhanced by means of methods characteristic for such sports. Exercises repeated in sets of jumps, runs and dynamic strength exercises induce high OI values [33].

The research on bone growth in children and adolescents in a large cohort of American children and adolescents (1554 girls and boys aged 6–16 years) found significant longitudinal effects of weight-bearing PA on bone mass accrual through all stages of pubertal development. It was found that self-reported weight-bearing PA contributed to a significantly higher increase in BMC in both genders [52].

Apart from PA, the study also analysed the dependence of BMD and BMC in childhood on birth parameters. The frequency of the individual birth weight (BW) categories was most favourable for track and field athletes. Furthermore, LBW was reported in almost half of the young swimmers and non-athletes, translating into worse BMD and BMC results compared to the group of track and field athletes with the lowest LBW frequency. Depending on PA, it was the birth weight that proved to be highly significantly related to the sT-score prox at age control.

Growing evidence suggests that LBW is associated with subnormal PBM [2,24]. Studies of young adults aged 18–27 years born with LBW indicate significantly lower total BMD and local BMD compared to their peers born with normal BW [2]. Furthermore, Wang et al. [53] observed that LBW was associated with the reduction of bone mass already at childhood age between five and 10 years, significantly reducing the chances to build optimal PBM. Our results of the study of boys confirm this suggestion, with swimmers and non-athletes characterized by a high frequency of LBW, which translated into a high frequency of low BMD in both forearm locations, especially in the proximal segment. PBM is considered to be the most important determinant of the development of osteoporosis and incidence of osteoporotic fractures in later stages of ontogenesis, which emphasizes the importance of an adequate increase in BMC in early childhood. Our study also demonstrated that boys with LBW have a significantly less favourable biological bone profile and lower bone mineralization in all analysed bone parameters. The subjects with normal and large birth weight showed significantly higher bone mineralization, and more favourable biological profiles.

In our research, length of breastfeeding was most often at the recommended levels in athletes, i.e., in the group with the best bone status. The study failed to show significant relationships between length of breastfeeding and bone status in boys. Similar to the results of the previous research [29]. Breastfeeding proved to be relevant only for the % age-matched prox and Z-score prox. However, it is worth emphasizing that the recommended length of breastfeeding in our study was much more frequent in the group with the best bone status, i.e., track and field athletes. Too short breastfeeding occurred in over 60% of boys from both groups with poorer bone mineralization.

Previous studies have shown contradictory results concerning both short-term and long-term effects of breastfeeding on BMC and BMD of children, adolescents and adults [29]. Some authors have demonstrated the positive effects of breastfeeding on BMC during childhood and puberty [54,55]. Jones et al. [25] found that children who had been breastfed longer than three months had higher BMC values in the eighth month of life compared to peers not breastfed at all or for shorter times. Furthermore, Molgaard et al. [26] observed a direct relationship between the length of breastfeeding and BMC in young adults aged 17 years. Some studies have indicated that early exposure to breast milk, even for a short period of time, may lead to changes in the programming of bone cells, resulting in higher bone mass in later life [56]. There are also results presented in available studies showing neither negative nor beneficial effects of breastfeeding on bone mass [57]. This issue requires further multifaceted research.

This study makes an important contribution to this area of research, since previously it was mainly demonstrated that the impact of swimming on the skeleton is less than the PA based on resistance exercises, but mainly in adult swimmers. In young boys training to swim, early detection of the risk of low BMD allows to take effective prophylactic measures and reduce the risk of osteopenia.

### Strengths and Limitation

The major strength of the study is a multi-factorial analysis of determinants of key bone parameters that offers the opportunity to assess the strength and direction of the effect of several important and diverse determinants rather than a single determinant on BMD and BMC. The analyses used in the study exclude indirect effects resulting from the interrelationships between the analysed predictors. Another strength of the present study is that a reliable and accurate research methodology was used. The research was conducted by a highly-qualified team with many years of research experience in the field. All data were collected using well selected and internationally recommended research tools. Data on bone parameters were collected by one highly-specialized expert, which excludes the existence of an intra-group error in bone parameter values. The research and reasoning were based on data of groups of boys with long training experience, who systematically trained distinctly different sports. This allows for drawing the conclusions about the links between the specific training on land and in water on bone parameters.

One of the study limitations is the relatively small yet sufficient size of the study group. It cannot provide a full representation of the population of Polish boys at this age, although it is satisfactory for drawing the conclusions concerning young physically active boys. Another limitation can be the lack of analogous data for young girls, but this was not the aim of this study. It would be advisable to extend the analysis to include BMD and BMC measurements at subsequent skeletal locations.

## 5. Conclusions

In this study, the type and level of PA proved to be an important determinant of bone parameters. We observed significant differences in BMD and BMC between groups of boys who are involved in track and field sports, which are characterized by exercises with a high osteogenic index (OI), and their peers training in water and those who are physically inactive.

The strongest links with bone parameters were found for the type of PA and birth weight. Regardless of birth weight, track and field athletes had the most advantageous bone parameters (mainly sT-score prox values). Swimmers with normal or LBW had less favourable sT-score prox values than non-athletes. The type of PA proved to be an important determinant of bone parameters.

## Figures and Tables

**Figure 1 ijerph-18-00245-f001:**
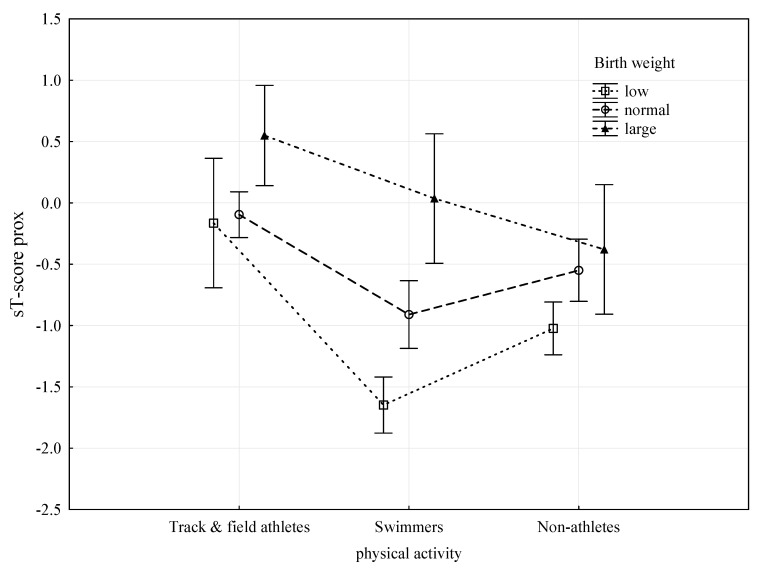
Relationships of PA category and birth weight with sT-score prox (ANOVA results).

**Figure 2 ijerph-18-00245-f002:**
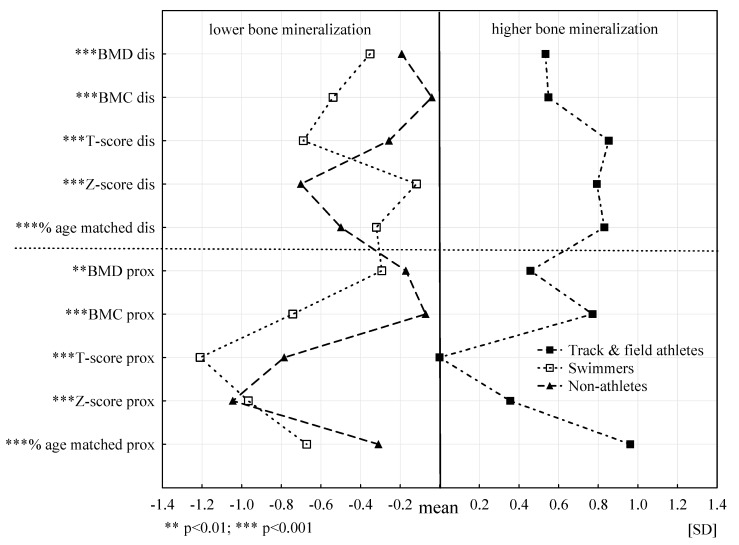
Biological profiles of the bone tissue mineralization state in three groups of PA.

**Figure 3 ijerph-18-00245-f003:**
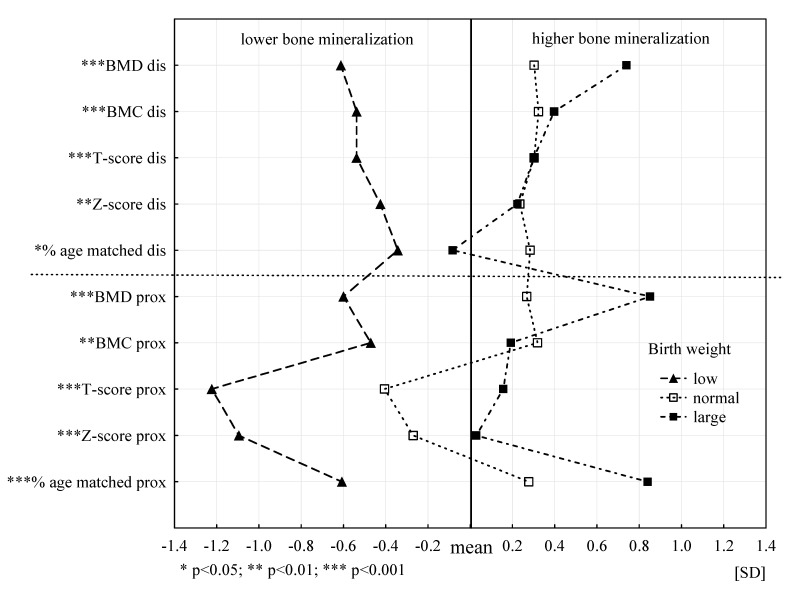
Biological profiles of the bone tissue mineralization status in three categories of birth weight.

**Table 1 ijerph-18-00245-t001:** Characteristics of study population (*n* = 96) (results of Kruskal-Wallis test and multiple comparisons of mean ranks).

Variables	Track & Field Athletes (*n* = 32)	Swimmers (*n* = 30)	Non-Athletes (*n* = 34)	F Test (*p*)
		Mean ± SD		
Age (years)	15.46 ± 1.19	16.18 ± 1.01	16.66 ± 0.72 ^#,^***	16.22 (0.001) ***
**Somatic and body composition**				
Weight (kg)	68.8 ± 6.4	63.8 ± 5.2 ^^,^*	73.2 ± 10.0 ~^,^***	20.70 (0.001) ***
Height (cm)	170.3 ± 6.1	173.9 ± 5.7	173.6 ± 7.4	4.66 (0.097)
BMI (kg/m^2^)	23.7 ± 1.7	21.1 ± 1.2 ^^,^***	24.3 ± 3.4 ~^,^***	30.65 (0.001) ***
Fat (%)	15.8 ± 2.1	12.8 ± 1.5 ^^,^***	16.8 ± 4.1 ~^,^***	31.41 (0.001) ***
FM (kg)	22.9 ± 2.2	20.1 ± 1.7 ^^,^***	22.8 ± 3.4 ~^,^**	21.40 (0.001) ***
FFM (kg)	45.8 ± 6.8	43.7 ± 5.6	50.4 ± 9.3 ~^,^**	11.39 (0.002) **
**Birth**				
Birth weight (g)	3121.9 ± 669.9	2635.5 ± 871.8	2535.14 ± 818.1 ^#^^,^*	8.64 (0.013) *
Length of breastfeeding (months)	9.2 ± 3.2	5.1 ± 4.8 ^^,^**	4.9 ± 4.6 ^#^^,^**	15.84 (0.001) ***
**Bone**				
BMD dis (g/cm^2^)	0.490 ± 0.106	0.388 ± 0.150 ^^,^***	0.411 ± 0.09 ^#^^,^*	16.52 (0.001) ***
BMC dis (g)	1.993 ± 0.389	1.391 ± 0.476 ^^,^***	1.658 ± 0.609 ^#^^,^*	20.95 (0.001) **
T-score dis	1.288 ± 1.341	−0.945 ± 0.874 ^^,^***	−0.356 ± 0.988 ^#^^,^***	41.39 (0.001) ***
Z-score dis	1.133 ± 1.194	−0.054 ± 0.505 ^^,^**	−0.803 ± 0.978 ^#^^,^***^,^ ~^,^**	45.26 (0.001) ***
% age matched dis	112.6 ± 16.7	90.9 ± 13.0 ^^,^***	87.1 ± 14.4 ^#^^,^***	39.11 (0.001) ***
BMD prox (g/cm^2^)	0.870 ± 0.086	0.730 ± 0.257 ^^,^***	0.755 ± 0.177 ^#^^,^**	25.48 (0.001) ***
BMC prox (g)	2.664 ± 0.580	1.705 ± 0.364 ^^,^***	2.143 ± 0.596 ^#^^,^**^,^ ~^,^*	33.63 (0.001) ***
T-score prox	−0.002 ± 1.027	−2.254 ± 1.276 ^^,^***	−1.464 ± 0.927 ^#^^,^***	42.18 (0.001) ***
Z-score prox	0.599 ± 1.034	−1.439 ± 0.752 ^^,^***	−1.580 ± 0.615 ^#^^,^***	49.82 (0.001) ***
% age matched prox	98.1 ± 9.7	75.5 ± 7.7 ^^,^***	78.7 ± 8.3 ^#^^,^***	53.70 (0.001) ***

Legend: BMI—body mass index; FM—fat mass; FFM—fat-free body mass; BMD—bone mineral density; BMC—bone mineral content; dis—distal, prox—proximal; F test—Ronald A. Fisher’s test; *p*—*p*-value, the levels of statistical significance: * *p* < 0.05; ** *p* < 0.01; *** *p* < 0.001. Markings used for results of multiple comparisons of mean ranks: ^ differences between Track & field athletes and Swimmers, ^#^ differences between Track & field athletes and Non-athletes, ~ differences between Swimmers and Non-athletes.

**Table 2 ijerph-18-00245-t002:** Assessment of the incidence of low BMD (osteopenia), low fat in body, low birth weight and not recommended length of breastfeeding (results of Chi-square test, *p*-value).

Variables	All (*n* = 96)	Track & Field Athletes (*n* = 32)	Swimmers (*n* = 30)	Non-Athletes (*n* = 34)
	%			
**T-Score dis**				
Low-osteopenia	26.0	0	53.3	26.5
Norm	74.0	100	46.7	73.5
**Chi-square test, *p***		29.36, 0.001		
**T-Score prox**				
Low-osteopenia	57.3	18.7	83.3	70.6
Norm	42.7	81.3	16.7	29.4
**Chi-square test, *p***		31.92, 0.001		
**Body fat**				
Low	0	0	0	0
Norm	89.6	93.7	100	76.5
Overweight	10.4	6.3	0	23.5
**Chi-square test, *p***		12.09, 0.002		
**Birth weight**				
Large for gestational age	11.5	15.6	10.0	8.8
Norm	50.0	75.0	36.7	38.2
Low	36.4	9.4	50.0	50.0
Very low	2.1	0	3.3	3.0
Extremely low	0	0	0	0
**Chi-square test, *p***		20.08, 0.003		
**Length of breastfeeding**				
Too short	54.2	28.1	66.7	67.7
Recommended	45.8	71.9	33.3	32.3
**Chi-square test, *p***		13.40, 0.001		

Legend: T-Score dis—T-Score distal; T-Score prox—T-Score proximal; *p*—*p*-value, level of statistical significance.

**Table 3 ijerph-18-00245-t003:** The strength of relationships of major determinants of biological bone mineralization status with all bone parameters (results of ANCOVA analyses, age-continuous variable).

	Distal Part			Proximal Part		
	Mean Square	F	*p*	Mean Square	F	*p*
	**BMD**			**BMD**		
age	0.0912	8.5464	0.0044	0.1435	5.3320	0.0233
PA	0.0411	3.8476	0.0250	0.0502	1.8655	0.1609
% fat	0.0125	1.1746	0.2814	0.0283	1.0508	0.3081
birth weight	0.0909	8.5209	0.0004	0.2377	8.8299	0.0003
breastfeeding	0.0049	0.4599	0.4995	0.0518	1.9234	0.1690
**F (*p*)**	6.54 (0.001)			6.11 (0.001)		
**R^2 adj.**	0.2897			0.2736		
	**BMC**			**BMC**		
age	0.7227	3.2956	0.0729	2.5085	9.9681	0.0023
PA	1.2226	5.5751	0.0052	5.0654	20.1289	0.0001
% fat	0.8136	3.7100	0.0573	0.2546	1.0117	0.3172
birth weight	0.9566	4.3621	0.0156	0.4770	1.8955	0.1563
breastfeeding	0.3195	1.4567	0.2307	0.3168	1.2589	0.2649
**F (*p*)**	6.40 (0.001)			10.31 (0.001)		
**R^2 adj.**	0.2848			0.4069		
	**T-score**			**T-score**		
age	12.6512	12.3716	0.0007	5.3535	7.3442	0.0081
PA	29.6370	28.9820	0.0001	19.3804	26.5869	0.0001
% fat	1.6325	1.5965	0.2097	0.0431	0.0590	0.8087
birth weight	2.3912	2.3384	0.1024	15.0226	20.6087	0.0001
breastfeeding	1.3514	1.3215	0.2534	2.5274	3.4673	0.0659
**F (*p*)**	14.58 (0.001)			24.62 (0.001)		
**R^2 adj.**	0.5004			0.6351		
	**Z-score**			**Z-score**		
age	7.7111	9.1404	0.0033	1.8828	3.4356	0.0672
PA	23.9460	28.3844	0.0001	24.5771	44.8469	0.0001
% fat	0.0140	0.0166	0.8977	0.0012	0.0022	0.9629
birth weight	0.7418	0.8793	0.4187	3.4629	6.3188	0.0027
breastfeeding	0.3290	0.3900	0.5339	2.6681	4.8687	0.0300
**F (*p*)**	12.10 (0.001)			28.50 (0.001)		
**R^2 adj.**	0.4498			0.6696		
	**% age matched**			**% age matched**		
age	1856.4875	8.9859	0.0035	248.9790	4.6291	0.0342
PA	5204.3858	25.1906	0.0001	2353.5956	43.7588	0.0001
% fat	0.0466	0.0002	0.9880	10.0730	0.1873	0.6662
birth weight	214.0252	1.0359	0.3592	389.1474	7.2352	0.0012
breastfeeding	17.7829	0.0861	0.7699	685.8800	12.7521	0.0006
**F (*p*)**	10.03 (0.001)			31.05 (0.001)		
**R^2 adj.**	0.3995			0.6889		

Legend: BMI—body mass index; FM—fat mass; FFM—fat-free body mass; BMD—bone mineral density; BMC—bone mineral content; F—Ronald A. Fisher’s test; *p*—*p*-value, level of statistical significance; PA—physical activity; R^2 adj.—the adjusted R-squared values of determination.

**Table 4 ijerph-18-00245-t004:** The strength of the relationships of the PA and birth weight with sT-score prox (the results of the two-way analysis of variance, PA and birth weight as determining variables, sT-score prox as the dependent variable).

Variables	Mean Square	F	*p*
PA	4.0773	19.2856	0.0001
Birth weight	3.5671	16.8724	0.0001
interaction	0.5122	2.4226	0.0542

Legend: F—Ronald A. Fisher’s test; *p*—*p*-value, level of statistical significance.

## Data Availability

The data presented in this study are available on request from the corresponding author.

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
