# Peer review of "Bone Mineral Density in Adolescent Boys: Cross-Sectional Observational Study"

_ijerph, 2020, doi:10.3390/ijerph18010245_

Round 1
Reviewer 1 Report
General
- The authors may consider specifying in the title and in the objective that this study is a cross-sectional observational study on the 3 groups of adolescents.
- Why was Kruskal-Wallis test used instead of ANOVA for continuous various able? Was it because of the violation to the normality test? Please specify in the methods.
- Authors may also consider specifying statistical test used in the caption for each table.
- Were pos-hoc tests performed to identify statistical difference between groups?
- [Table 2] Osteopenia/osteoporsis in adolescent is defined very differently compared to primary osteoporosis in older adults. The definition is not discussed in the introduction, nor specified in the materials and methods that I think should be supplemented. The result is making me skeptical that a rather high percentage of osteopenia is found in these rather active adolescents (normal, swimmers and runners), not including a sedentary group.
- Many of the definitions presented in table 2 were not specified in the materials and methods. g. osteopenia, body fat ranges for the low/norm/overweight, birth weight and cut-off values, etc.
- Table 3, please specify what is MS?
- Figure 1, line graph may not be the best representation of the data. There is no linear relationship between the 3 groups of subjects.
Minor
- Line 198, some of the abbreviations are not previously mentioned. g. “dis and % age matched dis”
- Table 1, commas (,) are used instead of periods (.) for decimal places. Please check if checks has specifications to follow.
- Table 2, The information provided is
Author Response
We would like to thank the Reviewer for her/his helpful and valuable comments. We have responded to these comments, and revised our manuscript accordingly. We hope that our article gained in clarity and correctness. We included all comments and the manuscript has been amended as specified in his remarks. Specific list of changes is given below.

Reviewer 2 Report
IJERPH 1022430
Thank you for the opportunity to review this manuscript. It is a well written manuscript and provides valuable information to establish prevention strategies in the child and adolescent population. This manuscript tries to s to assess bone mass (BMC) and bone mineralization (BMD) in young boys with different levels of physical activity and to evaluate the effects of body composition, birth weight and length of breastfeeding during infancy on bone parameters
Introduction,
I feel the authors have made a clear statement of the magnitude of the problem and its justification
Minor: please, consider add lines 66 and 67
Methods,
There are some aspects to review:
What type of study is it?
Could you modify levels of physical activity by types of physical activity? Throughout the manuscript they refer both concepts and I feel they are not using levels, but types. It can lead to confusion.
Could the authors add what they consider normal physical activity for this age? It is also necessary to have the criteria for inclusion and exclusion
Could the authors add the sample size?
Parents gave consent for their and their children's participation? Please add it
Results,
Please check the beginning of the results, as the first reference is a table.
Have the authors considered or is it possible to correct the FFM by height? This would help to understand the results
Please, could the authors add the cut-offs for each variable in Table 2
Lines 221 and 222, please write results
Lines 231 and 268, please change the symbol to the word
Table 3: Please use the same way of pointing out the levels of significance throughout the manuscript with arterisks and not in bold in this case.
Table 4: please add legend
Figures 2 and 3 are very useful, thank you very much
Discussion
I think it is well developed but some results should be discussed, such as lines 347-353
Conclusions,
Please write down the conclusions derived from the study and not those found by other authors.
References
Minor, review the format of the journal
All the best in your submission!
Author Response

(The authors gave the same response as above.)

Reviewer 3 Report
Thank you for the possibility to review the manuscript “Bone mineral density in adolescent boys: the effects of physical activity, birth factors, length of breastfeeding and body composition”. The study compares Polish boys with varied physical activity and observed a difference in bone parameters between the different groups.
Major comments
It needs to be clearly outlined what this study contributes to this field of research as it has been shown previously that the effect of swimming on the skeleton is less than impact based physical activities, please see e.g. reference: Gomez-Bruton A, Montero-Marín J, González-Agüero A, Gómez-Cabello A, García-Campayo J, Moreno LA, Casajús JA, Vicente-Rodríguez G. Swimming and peak bone mineral density: A systematic review and meta-analysis. J Sports Sci. 2018 Feb;36(4):365-377. doi: 10.1080/02640414.2017.1307440. Epub 2017 Apr 10. PMID: 28394711.
It also needs to be clarified in the paragraph regarding study population how the participants were identified. Was it by advertisement and if that was the case how and where? Are the participants representative for a larger group of boys or is it a very exclusive group of boys? The authors must also consider how the recruitment process of the participants influence the results.
It needs to be clearer in the methodology if the participants had performed any other sports previously that may affect the bone data? Please clarify. Please also specify the number of hours per week that the different groups use to perform their sports? Please clarify if any participant may perform other sports at the same time to the or not?
The study does not consider the difficulty of confounding and its effect on the results. A child born in a family of lower socioeconomic status, birth-weight may be influenced of the mother’s nutrition during pregnancy, may be breastfeed for shorter time and grow-up in a surrounding where physical activity is not prioritized, food may be lower in nutrition and other risk factors e.g. smoking may be prevalent. Which of these risk factors is the cause for lower bone parameters? These considerations need to be added to the discussion.
Why did the researchers not use medical transcripts of birth variables for e.g. birth weight, gestational length and maternal comorbidity? Please develop this reasoning as the data it is such an important part of the study, but it relies solely on self-reported data, but it should be possible to gain access to the source data. If not able to gain access, please state why. There is also no consideration regarding gestational gestational length at birth in the study and its effect on birth weight.
Lastly, why were the bone measurements performed on the arm and not at the spine or total body as recommended for children? See position statement: https://iscd.org/learn/official-positions/pediatric-positions/ and reference: Bachrach LK, Gordon CM; SECTION ON ENDOCRINOLOGY. Bone Densitometry in Children and Adolescents. Pediatrics. 2016 Oct;138(4):e20162398. doi: 10.1542/peds.2016-2398. PMID: 27669735.
Minor comments
Line 61: Please add further references as references 6,7 and 8 are all referring to Polish studies.
Line 70: Please use small letters for body mass index
Line 74: Please write out the abbreviation for BMC when it occurs the first time
Line 78: Please write out the abbreviation for PA when it occurs for the first time
Line 87: Please reformulate the sentence “Breastfeeding is widely considered a perfect way to feed infants.” as the formulation infers a subjective value on breastfeeding.
Lines 91-93: Please rephrase the sentences: “Interestingly, one study of adults showed the negative effect of too long breastfeeding time on bone mass in men. In women, breastfeeding did not affect bone mass [29]. “ as it is not clear whether the effect is on bone mass of the mother (woman) breastfeeding or the female child being breastfeed (also woman).
Lines 123-124:Please clarify what constitutes normal daily physical activity in the non-athletes group.
In the section regarding ethical approval, please clarify how the consent was achieved as all participants were minors at the time of the examination? Did their parents/legal guardians sign the consent form?
Line 132: Please state which research committee that approved the study.
Line 149: Please write out how the participants birth weights were compared to WHO standards, and as grouped in table 2, please state the definitions of the respective groups.
Line 151: Abbreviations such as BMC and BMD only need to be explained the first time they occur in the manuscript.
Line 198: Please write out the meaning of “dis” the first time it occurs. Presumably it is distal, but please clarify.
Table 1: Please write out in the abbreviations the meaning of “F”.
Lines 204-211: Please use double spacing.
Table 2: There are no p-values in the table, are there supposed to be? I presume this, as you are drawing conclusions from the table. If there are no participants in the extremely low birth weight, please write that out, as in the low birth weight group, or remove it from the table and write it in the text.
Table 3: Please write out all abbreviations used in the table.
Line 238: Please use small letters for birth.
Line 277: Is it “low body weight” or “low birth weight” that is meant in the sentence?
Line 282: Please clarify what “people” refers to, is it the participants in the study with normal or low birth weight?
Line 300: “The research”, does this refer to previous studies or the current study, please change the wording.
Lines 304-311: Please change this section. It is not necessary in the discussion to write detailed data about the exact levels of bone mineral density from another study, but rather use the other study to compare with the current study, what are the similarities/differences.
Lines 338- 340: According to figure 3 the bone mineralisation status was worse in all parameters in the low birth weight not only the swimmers and non-athletes, why is this section not referring to all participants?
Line 392: Repetition of abbreviation for osteogenic impact (OI).
Lines 394-396: The last two sentences of the conclusion should be removed as they have not been the scope of the article.
Author Response

(The authors gave the same response as above.)

Round 2
Reviewer 2 Report
I believe that the manuscript has improved significantly
Author Response
We would like to thank the Reviewer for his valuable commentd which has helped us to improve our manuscript.
Reviewer 3 Report
Thank you for the reviewed version of the manuscript.
The manuscript has improved but I have still some remaining comments from my first review that, in my opinion, have not been clarified enough or changed.
Major comments
It is still unclear in the text how the study participants were identified. It is said that the boys were invited, but how were the boys identified? Was it through advertisements or how was the recruitment process? This is important regarding the “healthy cohort effect” and needs to be clarified in the material and methods section.
I find the authors’ comment, that it is space capacity that makes it impossible to address the question of confounding, very unsatisfying. A short paragraph in the discussion regarding this would have been satisfactory.
Minor comments
Table 2 still has no p-values and there is no legend with abbreviations.
Line 69: Body mass index is still with capital letters, please change to small letters.
There is still inconsistent use of abbreviations and its subsequent explanation. The authors repeat the full wording in several cases when the abbreviation has already been shown previously this applies e.g. for physical activity (line 76 already shown in line 61); bone mineral content (line 86 already shown in line 73), and for other abbreviations such as BMD. Please check this in the manuscript so that the wording of the abbreviation only is shown the first time and thereafter the abbreviation is used.
It is still not clear which research ethics committee that approved the research i.e. which university or regional committee.
There is still no definition in the text of what constitutes normal birth weight, low birth weight, very low birth weight etc. Please add definitions of this.
Line 234: It is not clear if the sentence refers to all participants or just a subgroup as the percentage of 67% corresponds to the respective groups of swimmers and non-athletes, but not to all participants where 54,2% were breastfeed shorter than 6 months. Please clarify.
Author Response
We would like to thank the Reviewer for his valuable comments which has helped us to improve our manuscript.
